Review

 

**Subject Area:**
biochemistry/microbiology/molecular biology

chromosome organization, chromosome segregation, chromosome maintenance, ParA–ParB–*parS*, spreading, SMC

**Author for correspondence:**
Tung B. K. Le
e-mail: tung.le@jic.ac.uk

# Bacterial chromosome segregation by the ParABS system

Adam S. B. Jalal and Tung B. K. Le

Department of Molecular Microbiology, John Innes Centre, Norwich NR4 7UH, United Kingdom

ASBJ, 0000-0001-7794-8834; TBKL, 0000-0003-4764-8851

Proper chromosome segregation during cell division is essential in all domains of life. In the majority of bacterial species, faithful chromosome segregation is mediated by the tripartite ParABS system, consisting of an ATPase protein ParA, a CTPase and DNA-binding protein ParB, and a centromere-like *parS* site. The *parS* site is most often located near the origin of replication and is segregated first after chromosome replication. ParB nucleates on *parS* before binding to adjacent non-specific DNA to form a multimeric nucleoprotein complex. ParA interacts with ParB to drive the higher-order ParB–DNA complex, and hence the replicating chromosomes, to each daughter cell. Here, we review the various models for the formation of the ParABS complex and describe its role in segregating the origin-proximal region of the chromosome. Additionally, we discuss outstanding questions and challenges in understanding bacterial chromosome segregation.

## 1. Introduction

Faithful chromosome segregation is essential to ensure each daughter cell inherits a full copy of the genetic information of the parent. Chromosome segregation is not a trivial process, especially in bacteria, because DNA must be maintained in a compacted state to fit within the limited volume of the cells, and chromosome segregation often occurs concomitantly with DNA replication rather than being separated temporally, as in eukaryotes. Bacterial chromosome segregation can be divided into multiple overlapping steps: (i) segregation of DNA proximal to the origin of replication, (ii) segregation of the bulk of the chromosome, and (iii) segregation of DNA near the terminus of replication. In this review, we focus on progress towards understanding the molecular basis for segregating the origin-proximal region, specifically by the tripartite ParA–ParB–*parS* system.

The *par* locus was first discovered in low-copy-number plasmids, and was shown to be essential for their stable inheritance [1–4]. A functionally equivalent *par* locus was later found to be important for chromosome segregation in *Bacillus subtilis* [5–7]. In *Caulobacter crescentus*, *Hyphomonas neptunium* and *Myxococcus xanthus*, genes encoded in the *par* locus (ParABS) were found to be essential for cell viability [8–11], whereas in other bacterial species engineered strains lacking ParABS were viable but had an elevated number of anucleate cells owing to defects in chromosome segregation [12–27]. A comparative genomic study suggested that the chromosomal ParABS system is conserved in two-thirds of bacterial species [28]. In most bacteria, one or multiple *parS* sites are commonly found near the origin of replication [28]. The *parS* site is the first DNA locus to be segregated after chromosome replication [7,11,13,29]. ParB is a DNA-binding protein that nucleates on *parS* to recruit additional ParB molecules to adjacent non-specific DNA to form a network of protein–DNA complexes [30]. The ParB–DNA nucleoprotein complex stimulates the ATPase activity of ParA, creating a gradient of ParA–ATP that drives the movement of the origin-proximal region of the chromosome (and subsequently, the whole chromosome) along this gradient to the opposite pole of the cell [31–39]. ParB also recruits the structural maintenance of chromosome (SMC) complex onto the chromosome to

reduce DNA entanglement, thereby promoting the individualization of replicated chromosomes [16,40–46].

Since the discovery of the ParABS system over 35 years ago, tremendous progress has been made towards answering some of the key questions about how this system works:

— How does ParB recruit tens to hundreds more ParB proteins to assemble a higher-order nucleoprotein complex? (Discussed in §2.)
— What is the molecular mechanism of ParA-mediated DNA segregation? (Discussed in §3.)
— How does ParB recruits SMC and other protein partners to coordinate chromosome segregation with chromosome organization? (Discussed in §4.)
— How does evolution shape factors that are involved in bacterial chromosome segregation and maintenance? (Discussed in §5.)

In this review, we summarize recent progress and compare the competing models for addressing these key questions, before highlighting outstanding questions and challenges for fully understanding the ParABS system and chromosome segregation in bacteria.

# 2. ParB–parS interaction and the assembly of a higher-order nucleoprotein complex

ParB binding to parS nucleates the recruitment of additional ParB molecules which associate with neighbouring DNA, a process known as spreading, to form a higher-order ParB-DNA nucleoprotein complex [30]. The purpose of this higher-order complex, whether to strengthen the physical link between DNA and ParA or to provide a specific DNA topology to facilitate DNA segregation, is still under debate. However, since bacterial strains harbouring nucleation competent but spreading-defective mutants of parB are either unviable or have elevated number of anucleate cells, it is clear that a higher-order nucleoprotein complex is a prerequisite for faithful chromosome segregation [7,47–50]. In this section, we describe and discuss the current and emerging models for the assembly of this essential nucleoprotein complex.

## 2.1. Domain organization and shared features of chromosomal ParB protein family

Chromosomal ParB proteins share a common domain architecture, consisting of an N-terminal domain (NTD), a central DNA-binding domain (DBD) and a C-terminal domain (CTD) (figure 1a). A highly conserved arginine-rich motif (GERRxRA) resides in the NTD and mediates protein–protein and protein–ligand interactions [30,51,52] (figure 1a). The DBD contains a helix–turn–helix motif that enables ParB to nucleate on parS specifically [30]. The CTD, which is the least conserved domain among ParB homologs, contains a leucine zipper motif that allows ParB to homodimerize [30] (figure 1a). The CTD of Bacillus subtilis ParB also has a lysine-rich amino acid patch that provides additional non-specific DNA-binding and DNA condensation activities [53]. Currently, the structure of a full-length chromosomal ParB is not available. The flexibility of ParB, endowed by amino acid linkers that connect consecutive domains, has hindered the effort to crystallize

and solve the structure of a full-length protein. Nevertheless, structure-function insights have been gained from X-ray crystallography/NMR studies using a single-domain or domain-truncated variants of ParB from various bacterial species [52–58]. Structural comparisons suggested that ParB, especially its NTD, can adopt multiple alternative conformations that might facilitate the assembly of a higher-order nucleoprotein complex.

Four models have been proposed relating to the assembly of a higher-order ParB–DNA nucleoprotein complex. Here we assess the evidence for and against each model.

## 2.2. Model 1—one-dimensional filamentation of ParB

The earliest evidence of a higher-order ParB-DNA nucleoprotein complex came from studies of a plasmid-borne ParB. Overexpression of an F-plasmid ParB protein (ParB$_F$ or SopB) was observed to repress the expression of antibiotic resistance genes several kilobases away from the parS (sopC) site on the plasmid [59]. Moreover, ParB$_F$ overexpression also prevents DNA gyrase and restriction enzyme access to DNA regions neighbouring the parS site [59]. Similarly, a P1-plasmid ParB (ParB$_{P1}$) also silences the expression of genes adjacent to parS in both directions for several kilobases, with the efficiency of gene silencing decreasing as the genomic distance from parS increases [60]. A direct association of ParB$_{P1}$ with the silenced DNA was demonstrated by chromatin immunoprecipitation PCR (ChIP-PCR) assay [60]. Based on these observations, it was proposed the growth of a filament of ParB proteins nucleated at parS and then spread outward to neighbouring DNA (figure 1b). This model was further supported by the observation that a site-specific DNA-binding protein, RepA, could attenuate the ParB$_{P1}$-mediated gene silencing effect, presumably by acting as a roadblock to partially stop the filamentation of ParB [60] (figure 1b). Multiple chromosomal ParBs have subsequently been observed by ChIP-chip/seq to associate with an extended DNA region beyond parS [13,16,19,44,47,48,61,62], hence chromosomal ParBs were also thought to oligomerize to form a nucleoprotein filament. The highly conserved arginine-rich patch (GERRxRA) at the NTD has been implicated in mediating ParB filamentation, as mutations in this region impair the ability of ParB to associate extensively with DNA beyond parS [47–49,62,63]. This early model of ParB spreading is straightforward and attractive; however, later studies have argued that the intracellular concentration of ParB is too low to support such an extensive one-dimensional filamentation in vivo [62,64]. Moreover, at native expression levels, B. subtilis ParB (Spo0 J) does not silence genes adjacent to parS [48], suggesting that the ParB-DNA nucleoprotein complex might be more dynamic than can be explained by the one-dimensional filamentation model.

## 2.3. Model 2—bridging and condensing DNA

A combination of quantitative immunoblotting and immuno-fluorescence microscopy approaches led to the estimate that approximately 20 ParB dimers are associated with each parS site in B. subtilis, allowing for maximally approximately 500 bp of DNA to be covered by a continuous filament of ParB [62]. This is substantially lower than the approximately 10–20 kb of ParB-bound DNA observed by ChIP-chip [48,61], arguing against the one-dimensional filamentation model. Instead, a new model was proposed based on the observation that

royalsocietypublishing.org/journal/rsob    Open Biol. 10: 200097

royalsocietypublishing.org/journal/rsob    Open Biol. **10**: 200097

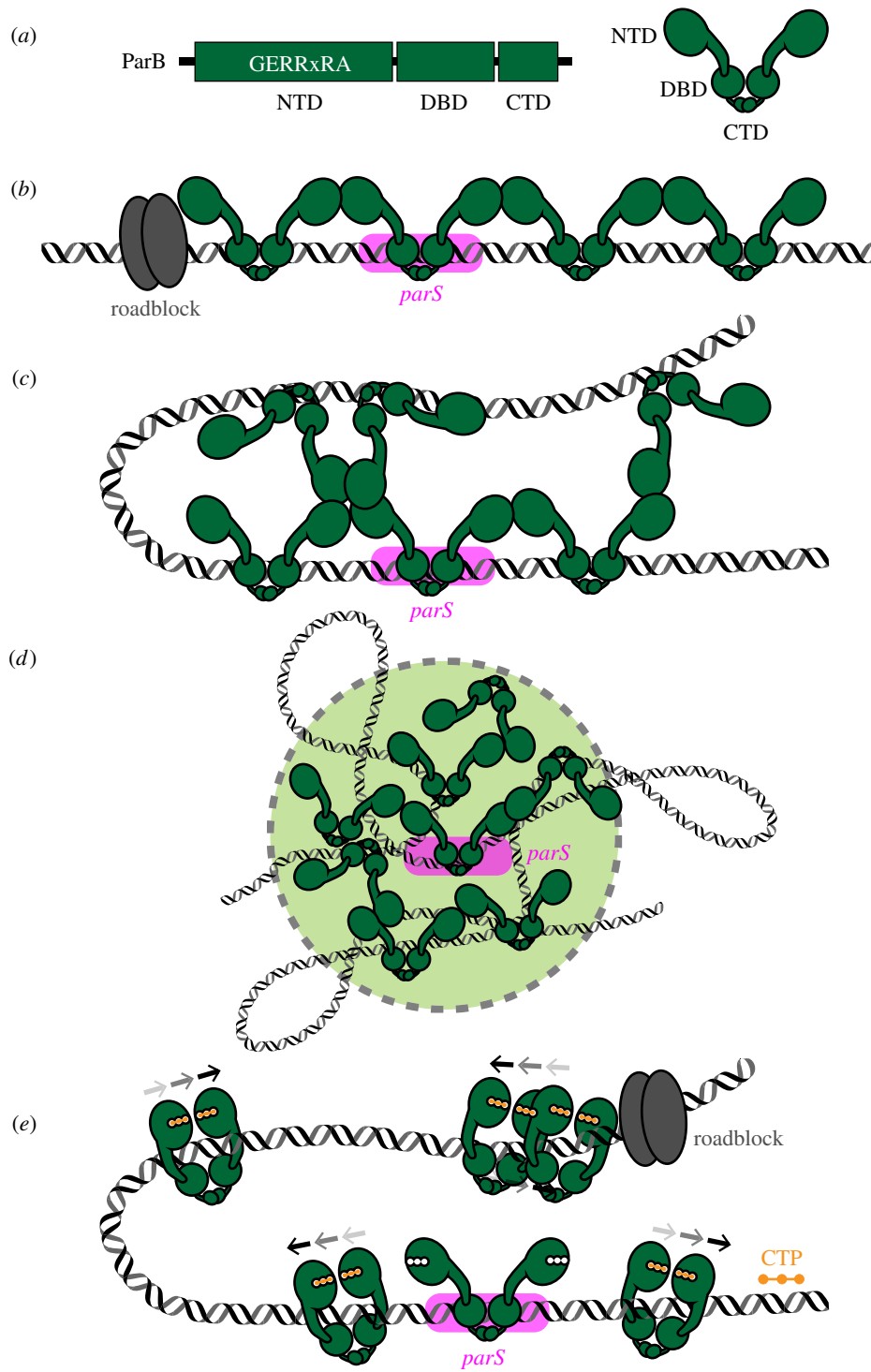

**Figure 1.** The assembly of a higher-order ParB–DNA nucleoprotein complex. (*a*) Chromosomal ParB proteins share a common domain architecture, consisting of an N-terminal domain (NTD), a central DNA-binding domain (DBD) and a C-terminal domain (CTD). The NTD harbours a conserved arginine-rich motif (GERRxRA) that mediates ParB–ParB and ParB–cytidine triphosphate (CTP) interactions. (*b*) Model 1: ParB spreading by a one-dimensional filamentation. (*c*) Model 2: ParB spreading by bridging and condensing DNA. (*d*) Model 3: ParB spreading by caging DNA. (*e*) Model 4: ParB spreading by sliding on DNA. ParB switches from an open to a closed clamp upon binding to CTP (orange). ParB and *parS* are coloured green and magenta, respectively. The arrows above the ParB–CTP complexes (*e*) indicate their progressive sliding on DNA. A tight DNA-binding protein (grey) can unidirectionally block the one-dimensional filamentation or the sliding of ParB on DNA.

*B. subtilis* ParB can bridge different segments of DNA (figure 1*c*). In a single-molecule microscopy-based assay, bacteriophage λ DNA (approx. 50 kb) was tethered at one end to a microscope slide and stretched out by a buffer flow. The introduction of purified *B. subtilis* ParB compacted the flow-extended DNA, demonstrating that ParB can form bridges and condense bound DNA [62]. Moreover, mutations in the arginine-rich patch which eliminate the extensive *in vivo* ChIP-seq profile of *B. subtilis* ParB also impair *in vitro* DNA-bridging activity [62].

ParB-mediated DNA bridging was also observed using magnetic-tweezers assays [65]. The additional non-specific DNA-binding activity owing to a surface-exposed lysine-rich patch at the CTD of *B. subtilis* ParB was found to be essential for this function [53,66]. Mutations in these lysine residues eliminate DNA bridging and condensation *in vitro* and reduce ParB-DNA nucleoprotein formation *in vivo*, as assessed by the less extensive ChIP-qPCR profile and by the dimmer and fuzzier appearance of fluorescently labelled ParB foci

royalsocietypublishing.org/journal/rsob    Open Biol. **10**: 200097

[53]. It is important to emphasize that interactions among NTD of *B. subtilis* ParB are also necessary for bridging DNA (figure 1*c*); neither CTD alone nor ParB with mutations at the arginine-rich patch (at the NTD) can condense DNA *in vitro* [53,66]. The non-specific DNA-binding activity of the CTD is thought to provide multiple anchors on DNA that can be brought spatially close together by the NTD–NTD interactions (figure 1*c*). Insights into the molecular mechanism of NTD–NTD interactions were provided by the co-crystal structure of a CTD-truncated *Helicobacter pylori* ParB in complex with a *parS* DNA duplex [54]. This structure shows *H. pylori* ParB interacting with an adjacent ParB on a pseudo-continuous DNA in the crystal lattice (in *cis* interactions or one-dimensional filamentation) and also with ParB on a disconnected DNA duplex (in *trans* interactions or three-dimensional bridging) (figure 1*c*), with the arginine-rich patch at the core of the NTD–NTD interaction interface [54]. By comparison with the *Thermus thermophilus* apo-ParB structure, it was proposed that the nucleation of ParB onto *parS* induces a conformational change at the NTD that exposes the arginine-rich patch for the NTD–NTD interactions [49,54,56].

In sum, it has been proposed that DNA-bridging activity allows a limited number of ParB molecules to bring regions of DNA that are several kilobases apart together in three-dimensional space to form a compacted nucleoprotein complex (figure 1*c*). Nevertheless, a computational modelling study has suggested that a combination of both one-dimensional filamentation and three-dimensional bridges are required to recreate the condensed ParB–DNA nucleoprotein complex observed *in vivo* [67]. Thus, while the DNA-bridging model is an important step towards understanding the assembly of the ParB–DNA nucleoprotein complex, it is unlikely to be the final say. The main caveat is that *B. subtilis* ParB can bridge to condense DNA *in vitro* regardless of the presence of *parS* [53,62,65]. This contradicts *in vivo* data showing *parS* is absolutely required for the clustering of fluorescently labelled ParB molecules into a tight focus [62,68]. Moreover, the lysine-rich patch (at the CTD of *B. subtilis* ParB) is not highly conserved; for example, ParB from *Caulobacter crescentus* lacks the equivalent lysine residues and does not bridge/condense DNA *in vitro* [55]. As such, it is not yet clear how prevalent DNA-bridging activity is among chromosomal ParB homologs.

## 2.4. Model 3—caging ParB and DNA

A model broadly similar to bridging and condensing DNA that aims to better explain the observed *parS*-dependent confinement of ParB *in vivo* has been proposed [69]. In this nucleation and caging model, the *parS* site acts as a ParB nucleation centre, while weak but synergistic protein–protein and protein–DNA interactions cage ParB spatially into a confined volume inside the cells [69] (figure 1*d*). Supporting this model, single-molecule super-resolution microscopy demonstrated that the binding of ParB$_F$ to *parS* results in a very high local concentration of protein *in vivo*, where greater than 90% of ParB$_F$ in the cell are confined in clusters at *parS* [69]. Similarly, the local concentration of *C. crescentus* ParB near *parS* has been estimated to reach approximately 500 µM (500 times more concentrated than typically used for *in vitro* experiments) [31]. Despite ParB$_F$ (or *C. crescentus* ParB) having expectedly low-affinity interactions with non-specific DNA, these interactions may occur stochastically at very high frequency,

especially at the extreme local concentration of ParB *in vivo*, to create a cage of dynamically exchanged ParB–DNA complexes (figure 1*d*). Fluorescence recovery after photo-bleaching (FRAP) experiments have shown that ParB$_F$ molecules rapidly exchange between different clusters, further highlighting the dynamic nature of cages of ParB-DNA *in vivo* [70]. The nucleation and caging model has also been shown applicable to the *Vibrio cholerae* chromosomal ParB–*parS* system, suggesting that this dynamic self-assembly mechanism might be conserved from plasmids to chromosomes [70].

## 2.5. Model 4—lateral sliding of a ParB–CTP clamp on DNA

Recent studies have uncovered a new cofactor of ParB [51,52]. Various plasmid- and chromosome-encoded ParB and ParB-like proteins have been found to bind and hydrolyse cytidine triphosphate (CTP) to cytidine di-phosphate (CDP) and inorganic phosphate [51,52,71]. A co-crystal structure showed CDP binding to the arginine-rich patch at the NTD of *B. subtilis* ParB (CTP was hydrolysed to CDP during crystallization) [52]. At the same time, another co-crystal structure showed a *M. xanthus* ParB-like protein (PadC) in complex with CTP [51]. CTP (or CDP) is sandwiched between two NTDs, thus promoting a new NTD self-dimerization interface that has not been observed previously [51,52]. Employing site-specific cross-linking assays and single-molecule imaging, it was demonstrated that CTP-induced self-dimerization creates a clamp-like ParB that entraps DNA within its central cavity [52] (figure 1*e*). A comparison between the *B. subtilis* ParB–CDP structure and the *H. pylori* ParB–*parS* structure suggested that CTP binding induces a conformational change at the central DNA-binding domain that is incompatible with *parS* binding [52,54]. Studies with *C. crescentus* and *M. xanthus* ParBs further showed that CTP binding reduces ParB nucleation at *parS* and/or liberates pre-bound ParB from *parS* [51,71], thereby facilitating the escape of ParB from a high-affinity nucleation site to a low-affinity neighbouring DNA. Therefore, CTP probably serves to switch ParB from a nucleating to a sliding mode (figure 1*e*). Overall, it was suggested that ParB clamp can self-load at *parS*, without the need of a dedicated loading factor, and spreads by sliding to the neighbouring DNA while still entrapping DNA [52,71] (figure 1*e*). The interpretation of a sliding ParB–CTP clamp on DNA is further backed up by several lines of evidence: (i) tight DNA-binding proteins, such as a catalytic-dead EcoRI (E111Q) variant or TetR, can block the spreading of *B. subtilis* and *C. crescentus* ParB–CTP on DNA *in vitro* [52,71] (figure 1*e*), and (ii) *C. crescentus* ParB only accumulates on DNA that has both ends blocked (by a bulky biotin-streptavidin complex) to prevent a run-off [71]. However, it is not yet clear whether the translocation of ParB–CTP on DNA is entirely a passive one-dimensional diffusion process or whether it is facilitated by unknown interactions between the protein and DNA. CTP hydrolysis is unlikely to provide energy for ParB translocation since its hydrolysis rate is extremely low, ranging from approximately 3 to approximately 36 CTP molecules per hour [51,52,71]. Moreover, ParB in complex with a non-hydrolysable CTP$\gamma$S analog can still self-load and accumulate on DNA, albeit with a reduced stability [52,71]. It has been speculated that CTP hydrolysis might contribute to recycling of ParB

between the nucleation and translocation modes [52,71]. Mutant proteins (N112S and N172A of *B. subtilis* and *M. xanthus* ParB, respectively), which bind CTP but are deficient for hydrolysis, fail to form tight foci inside the cells [40,51,52]; however, this is weak evidence for the *in vivo* role of CTP hydrolysis since *B. subtilis* ParB (N112S) is already impaired at forming a protein clamp [52]. A better understanding of the CTPase mechanism that enables the design of a mutation at the catalytic site to eliminate CTP hydrolysis while allowing NTD self-dimerization is likely to provide a key insight into the role of CTP hydrolysis.

## 2.6. Reconciliation of different models: outstanding questions and challenges

The unexpected finding of the ParB–CTP interaction has fundamentally changed thinking on the assembly of a higher-order nucleoprotein complex and bacterial chromosome segregation by the ParABS system. But does the 'ParB spreading by sliding' model supersede previously proposed models? It is too early to answer this question adequately, given that many mechanistic details are still missing. For example, an alternative view has been proposed wherein *parS* binding stimulates the CTPase activity to switch *M. xanthus* ParB from a CTP-bound closed conformation to an apo/CDP-bound open conformation, liberating the NTD to engage in DNA-bridging/caging interactions [51]. It is possible that there are two different modes of action of ParB inside the cells: one for bridging/caging DNA together, and another for the lateral sliding of ParB on DNA. Investigating the relative contribution of the two different modes of action to chromosome segregation, especially *in vivo*, is an important challenge. Some of other immediate questions to which answers can help refine or reconcile different models include:

— How dynamic is the ParB clamp opening and closing when bound to *parS* and/or to CTP?
— Can the ParB clamp entrap two or more DNA segments together [52], thereby contributing to DNA bridging and condensation?
— What is the mechanism of CTP hydrolysis?
— Does the translocation of ParB supercoil DNA, thereby compacting *parS*-proximal DNA?
— Is there a variation in CTP-binding affinity and CTP hydrolysis rate among ParB orthologs, and how does this natural variation impact chromosome segregation in different bacterial species?

Whether CTP plays a regulatory role in chromosome segregation, in addition to being a co-factor of ParB, is also unknown. The concentration of nucleoside triphosphate (NTP) ranges from approximately 0.3 to approximately 3 mM inside bacterial cells [72]. Their concentrations can decrease by ~tenfold as cells enter the stationary phase [72] but it is unlikely to impact ParB–CTP binding significantly. Indeed, foci of a fluorescently tagged ParB do not disappear when *C. crescentus* cells enter the stationary phase or during starvation [73]. For these reasons, we speculate that the assembly of ParB–DNA nucleoprotein complex is not regulated by varying the intracellular concentration of CTP. However, there is a formal possibility that other NTP-related small molecules, whose diversity has only been realized recently [74],

could have a regulatory impact. Future work will undoubtedly continue to provide important new insights into the assembly of the ParB–DNA nucleoprotein complex and its roles in chromosome segregation.

## 3. ParB–DNA interaction with ParA and segregation of the origin-proximal chromosomal region

ParA is a deviant Walker A ATPase protein [75] that enables a directional movement of ParB-bound DNA. Early studies of plasmid and chromosome segregation proposed a mechanism for DNA-pulling by either a linear or a helical ParA filament [76–83], akin to the mitotic spindle apparatus in eukaryotes. According to this model, ParA–ATP polymerizes into a filamentous structure along the cell length, with the edge of the filament capturing the ParB–DNA nucleoprotein complex. ParB binds ParA and stimulates its ATPase activity to hydrolyse ATP, thereby depolymerizing the ParA filament and concomitantly pulling the ParB–DNA complex (hence, the plasmid/chromosome) along the retracting filament to the opposite cell pole [76,83,84]. While purified ParA from various bacterial species could self-aggregate into filament-like structures in the presence of ATP/ADP [34,76,79,82,85–91], no such continuous polymer was seen in recent co-crystal structures of ParA with DNA, even at the high concentration of protein and DNA used to generate crystals. Furthermore, the spatial distribution of an F-plasmid ParA and *C. crescentus* ParA *in vivo* is inconsistent with a continuous filamentous structure, instead they form small patches or a cloud-like gradient of sparsely distributed molecules inside the cells, as observed by super-resolution microscopy [31,37]. As such, it is uncertain whether a DNA-pulling mechanism by a ParA filament is operating *in vivo*.

It has been proposed that a ParA filament is not necessary for DNA segregation, and that a diffusion-ratchet mechanism can also explain the directional movement of segregating DNA [33,35,36,38,92,93] (figure 2*a,b*). In this model, ParA binds ATP to homodimerize and to associate with non-specific DNA. X-ray crystallographic and hydrogen/deuterium exchange mass spectrometry analysis of ParA with nucleotides and DNA have revealed the dimerization interface and a multifaced DNA-binding surface [94–96]. ParB, via its N-terminal peptide, binds ParA directly and stimulates the ATPase activity of ParA, thereby dissociating ParA dimer into individual monomers that no longer bind DNA [34,36,97,98] (figure 2*a*). This stimulation in the ATPase activity creates a local gradient of ParA–ATP with the least DNA-bound ParA–ATP near the ParB–DNA complex (figure 2*a*). The ParB–DNA complex then diffuses up the gradient, by Brownian motion, to rebind ParA–ATP, resulting in a net movement of the ParB-anchored DNA (figure 2*a*). The initial movement of the ParB–DNA complex in one chosen direction enforces the continued movement in the same direction, resulting in a long-range directional movement of the DNA (figure 2*a,b*). The released monomeric apo–ParA/ParA–ADP can rebind ATP to homodimerize and later regains its non-specific DNA-binding activity (figure 2*a*). It is worth noting that the released apo–ParA/ParA–ADP can rebind ATP but cannot immediately bind DNA until a transition occurs in the ParA–ATP structure (figure 2*a*); this transitional state presumably introduces a time delay

royalsocietypublishing.org/journal/rsob   Open Biol. **10**: 200097

royalsocietypublishing.org/journal/rsob    Open Biol. **10**: 200097

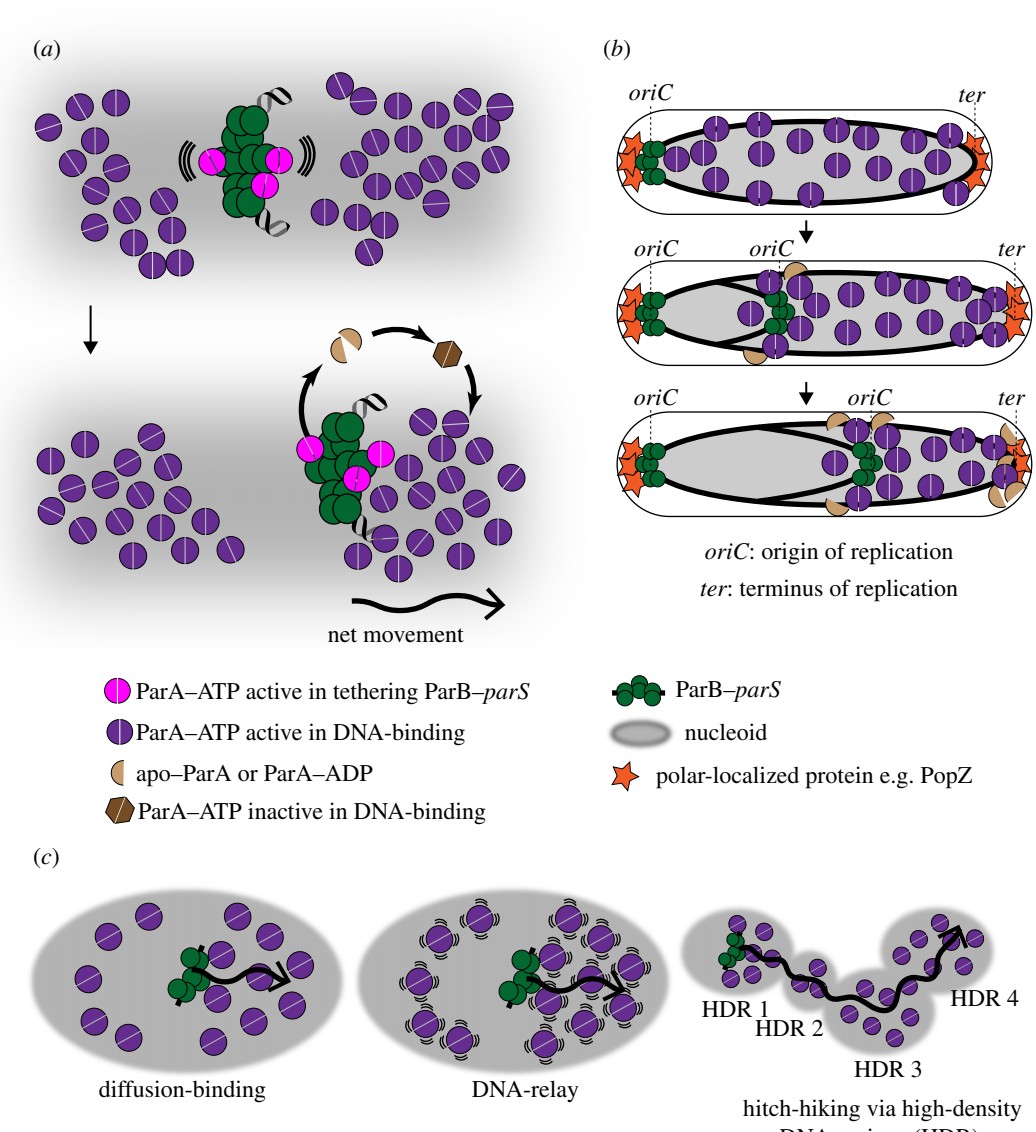

**Figure 2.** ParA drives the movement of ParB-bound DNA to segregate plasmids and chromosomes. (*a*) A diffusion-ratchet model for ParA-mediated transport of ParB-bound DNA. A ParB–DNA complex (green) interacts with ParA–ATP (violet) to tether to the nucleoid (grey), and to stimulate the ATPase activity of ParA. ParA–ATP dimers (violet) bind the nucleoid non-specifically. After ATP hydrolysis, monomers of apo–ParA/ParA–ADP (light brown) no longer bind DNA, thus creating a zone of depletion of ParA–ATP surrounding the ParB–DNA complex. By thermal fluctuation (wavy lines), the ParB–DNA complex moves to the edge of the zone of depletion to rebind ParA–ATP. The initial movement of the ParB–DNA complex in one chosen direction enforces the continued movement in the same direction, resulting in a long-range directional movement of the DNA (see *b*). The released apo–ParA/ParA–ADP (light brown) rebinds ATP but cannot immediately bind DNA (the dark brown hexagon) until a transition occurs in the ParA–ATP structure. (*b*) The segregation of the origin-proximal region of the chromosome by the ParABS system. For example, in *C. crescentus*, one ParB–DNA complex remains at the pole after chromosome replication, while the other moves along the gradient of ParA–ATP, via the diffusion-ratchet mechanism, to the opposite cell pole. The polarly localized proteins (e.g. PopZ, orange) contribute to maintaining the ParA–ATP gradient by sequestering apo–ParA/ParA–ADP away from the nucleoid and to regenerate them at the pole. (*c*) Other variations of the diffusion-ratchet model have been proposed to include an element of DNA elasticity (i.e. the DNA-relay model) or high-density DNA regions (HDR) (i.e. the hitch-hiking model). A wavy arrow indicates the directional movement of the partition complex.

mechanism to ensure the existence of a ParA–ATP gradient surrounding the ParB–DNA complex [38,94]. Without this delay, regenerated ParA–ATP will instantly rebind DNA in the same location, thus dissipating the gradient. Other organism-specific factors, for example, the polarly localized proteins PopZ and TipN in *C. crescentus*, may also contribute to maintain the ParA–ATP gradient by sequestering apo–ParA/ParA–ADP away from the nucleoid and to regenerate ParA–ATP only at the cell pole [99,100] (figure 2*b*). Based on computational modelling it has been argued that the short-range diffusion of a ParB–DNA complex up the gradient of ParA–ATP might not be sufficient to explain a robust uni-directional segregation of chromosome towards the new cell

pole (figure 2*b*) and that the diffusion-ratchet model should be extended to incorporate a component of DNA elasticity. In this model, DNA-bound ParA–ATP complexes can harness the elastic dynamics of the chromosome to relay the partition complex over a long distance from one DNA location to another [31] (figure 2*c*). Similarly, it has also been proposed that partition complexes can also hitchhike from one high-density DNA region to another on the chromosome to move the ParB-bound DNA progressively [37] (figure 2*c*). High-density DNA regions have been observed in *B. subtilis* and *Escherichia coli* by super-resolution microscopy and may represent highly compacted domains of the chromosomes [37,101]. The preferred association of ParA–ATP with high-density DNA

regions, via its non-specific DNA-binding activity, might create the required directional bias in the movement of the ParABS complex (figure 2*c*).

The diffusion-ratchet model emphasizes the crucial role of ParB in stimulating the ATPase activity of ParA to create the ParA–ATP gradient. However, an alternative view on the ATPase-stimulating role of ParB, at least for the F-plasmid ParAB system (SopAB) suggested that the stimulation of ParA$_F$ ATPase activity mainly serves to spatially separate F-plasmid clusters following replication and to prevent them from re-forming later [102]. The directional movement of replicated F plasmids might depend on a basal ATPase activity of ParA$_F$ but does not need further stimulation by ParB$_F$ [102]. Finally, the recent discovery of CTP as a cofactor of both plasmid- and chromosome-encoded ParB raises many important questions. Does ParB–CTP further stimulate the ATPase activity of ParA, and conversely, does ParA accelerate the CTP hydrolysis rate of ParB? Early evidence suggested that CTP can modulate ParA–ParB interaction; mutations at the CTP-binding pocket of a ParB-like protein PadC were shown to impair PadC–ParA binding *in vitro* (i.e. ParA preferentially binds to PadC–CTP, rather than to apo-PadC, and gave rise to aberrant ParA localization patterns *in vivo* [51]). Future works, especially with the canonical ParABS system, will provide important insights to refine current models for the ParA-directed DNA segregation.

## 4. The ParB-DNA and SMC coordinate chromosome segregation with chromosome organization

In addition to its role in DNA segregation, ParB also participates in other biological processes such as chromosome organization, nucleoid occlusion, regulation of DNA replication initiation and regulation of gene expression [16,24,40,41, 100,103–113]. The wide range of ParB-interacting partners reflects (i) the central role of the ParB-DNA nucleoprotein as a hub to couple chromosome segregation with other biological processes and (ii) the capacity of ParB to evolve additional functions. For a further discussion, we refer the reader to recent reviews [114,115]. In this section, we instead focus on the interaction between ParB and the SMC complex that is directly relevant to the segregation of the origin-proximal region of the chromosome.

A canonical bacterial SMC is composed of an ATPase domain (the head), a dimerization domain (the hinge) and an extended antiparallel coiled-coil region in the middle [116] (figure 3*a*). Two SMC monomers homodimerize together with the accessory proteins (ScpA and ScpB) to form a ring-like protein complex that can bring distal DNA segments close together spatially to organize the chromosome [116–118] (figure 3*a*). This entrapment of DNA has been shown for *B. subtilis* SMC [119] and for eukaryotic SMC homologs such as cohesin and condensin [120–123]. Application of chromosome conformation capture assays (Hi-C/ 3C-seq) to cells from a range of bacterial species lacking SMC have revealed a reduced interaction between opposite arms of the chromosome, suggesting that SMC entraps and tethers the two chromosome arms together [43,44,101,124–126] (figure 3*b*). SMC is recruited onto the chromosome by ParB at the origin-proximal *parS* sites [16,40–42,44] (figure 3*c*).

After loading, SMC redistributes directionally away from *parS* towards the replication terminus (*ter*) while maintaining the tethers between the *parS*-proximal regions of the chromosome arms [42,125] (figure 3*c*). Given that *parS* sites are often found near the origin of replication, *parS*-loaded SMCs preferably condense newly replicated DNA to package them into individual entities and away from each other (figure 3*b*). This DNA-unlinking activity is independent of topoisomerase IV, at least in *B. subtilis*, and might help to prevent catenation between replicated chromosomes at the replication fork or promote their resolution behind the fork [45,46]. If replicated chromosomes are not resolved, their entanglement might hinder movement of individual chromosomes to opposite cell poles by the ParABS system. In *C. crescentus*, segregation of origin-proximal DNA occurs in two steps; the duplicated origins are released from the pole and separate slightly from one another first before one of the origins is moved unidirectionally by ParABS to the opposite cell pole [127]. While the initial separation does not require ParA [127,128], it might be facilitated by the DNA-unlinking activity of SMC.

Precisely how SMC translocates on the chromosome is not yet clear; several models have been proposed, and we refer the reader to a recent review [129] for an in-depth discussion. How ParB loads SMC onto the chromosome is also not fully understood; the weak and transient interaction between ParB and SMC has made efforts to study their interactions by traditional methodologies (such as bacterial two-hybrid or co-immunoprecipitation) difficult [41,42,44]. However, it was suggested that DNA-bound ParB probably interacts directly with SMC to recruit it to the DNA [40]. Indeed, a ParB-interacting area has been identified in the neck region in between the ATPase head domain and the coiled coil of *B. subtilis* SMC [130], while mutations that eliminate SMC recruitment have been mapped onto the N-terminal domain of *B. subtilis* ParB [40,43]. Those same mutations also impair the ability of ParB to assemble into a higher-order nucleoprotein complex, hence it is tempting to speculate that either (i) a high local concentration of DNA-bound ParB is necessary to recruit sufficient SMC molecules or (ii) the DNA-bridging/ clamping activity of ParB ensures SMC entraps DNA correctly at the loading step. Future experiments, particularly a cell-free reconstitution of a ParB-dependent SMC recruitment and translocation, will provide further insights into the mechanism of actions of bacterial SMC and its contribution to chromosome segregation.

## 5. The evolution of the ParABS system and bacterial chromosome segregation

Research in multiple model species and an ever-increasing number of sequenced bacterial genomes has highlighted variations in the mechanism for bacterial chromosome segregation. Approximately 25% of bacterial species lack ParABS homologs entirely [28] and thus probably employ other systems to facilitate their chromosome segregation [131–133]. In some species, for example, *Streptococcus pneumoniae* or *Staphylococcus aureus*, only ParB–*parS* and SMC are present while a ParA homolog is missing [28]. Even in species with the canonical ParABS system, there exists a wide variation in the number of *parS* sites; for example, *Xanthomonas campestris* has a single *parS* site while *Streptomyces coelicolor* and *Listeria innocua* accumulated up to 20–23 *parS* sites near the origin of replication

royalsocietypublishing.org/journal/rsob    Open Biol. **10**: 200097

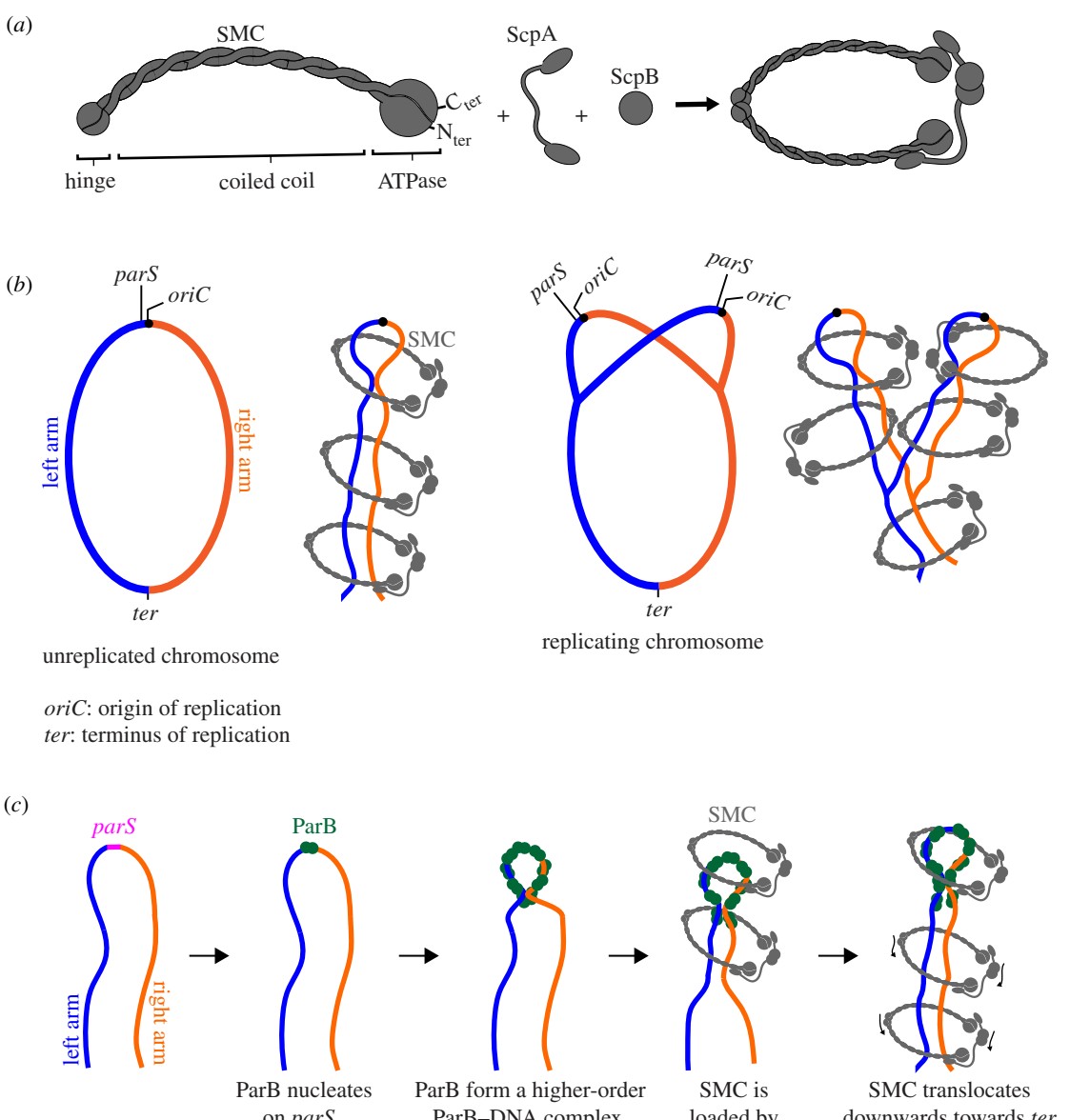

**Figure 3.** The ParB–DNA nucleoprotein complex recruits SMC to coordinate chromosome segregation and chromosome organization. (*a*) Components of the bacterial SMC complex. (*b*) SMC (grey) tethers the two arms (blue and orange) of a circular chromosome together. An SMC–ScpA–ScpB complex can either hold both the left and the right arm of the chromosome within its lumen or two SMC complexes, each encircles one chromosome arm, can handcuff to tether both chromosome arms together. For simplicity, only SMCs entrapping both chromosome arms are shown. SMC probably packages sister chromosomes into individual entities and away from each other, thus minimizing DNA entanglement between replicating chromosomes. (*c*) A schematic model of how SMC is loaded at *parS* by ParB and translocates on the chromosome towards the replication terminus (*ter*). For simplicity, alternative conformations of SMC (ring or rod) are not illustrated; the SMC complex is shown as a generic ring that entraps DNA. Schematic pictures are not drawn to scale.

[28,134]. Why is the number of *parS* sites variable when a single *parS* site is often sufficient for chromosome segregation [12,13,62]? How does this variation in the number of *parS* sites impact chromosome segregation in different bacterial species in their niches? Why do *parS* sites position closely on the genome and what drives their clustering over evolutionary time? For the last question, a transposon-based saturated insertion of a *parS* site on the *Pseudomonas aeruginosa* and *C. crescentus* chromosome offered some insights; it was discovered that the insertion of a *de novo* or a second *parS* site is only tolerable in approximately 600 kb region surrounding the native *parS* locus or the origin of replication without severely affecting cell fitness [13,47]. These results suggest a self-reinforcing mechanism for the expansion of the bacterial centromere region by restricting the multiplication of *parS* to a narrow region near the original site.

Another noteworthy example of the evolution of the ParABS system is the gene duplication and neo-functionalization event that generated a nucleoid occlusion factor (Noc) in Firmicutes [135–139]. Noc, a ParB-like protein, plays a role in preserving the integrity of the chromosome; it does so by preventing the cell division machinery from assembling in the vicinity of the segregating chromosome, which might be otherwise guillotined, thereby damaging the DNA [136,140,141]. An amphipathic helix is present at the N-terminus of Noc instead of the ParA–ATPase-stimulating peptide commonly found in ParB [141]. Mutations that perturbed the amphipathicity of this helix also eliminated the nucleoid occlusion function, while replacing the native helix with one from the hepatitis C virus protein NS4B restored the nucleoid occlusion activity [141,142]. A mutational event that resulted in the grafting of an amphipathic helix might

have been the evolutionary mechanism that once granted a novel function to a ParB protein [137,141]. Furthermore, in contrast to ParB, Noc does not bind *parS* but recognizes a different DNA-binding sequence called *NBS* (Noc-Binding Site) [140]. *NBS* differs from *parS* by only two bases but Noc and ParB recognize and bind them with exquisite specificity [140,143]. X-ray crystallography and systematic scanning mutagenesis identified a minimal set of just four amino acids that mediate ParB-*parS*/Noc-*NBS* binding specificity [143]. Deep mutational scanning of these four specificity residues enabled an *in silico* reconstitution of possible evolutionary paths that reprogramed DNA-binding specificity from *parS* to *NBS* [143]. A small number of required mutations and the large number of mutational paths to reprogram DNA-binding specificity illustrates the evolvability of the ParABS system.

The existence of various ParA homologs with diverse functions is also intriguing. In *Rhodobacter sphaeroides*, an orphan ParA-like protein (PpfA) uses non-specific nucleoid binding to separate cytoplasmic clusters of chemotaxis proteins [144]. Similar to the canonical ParABS system, the ATPase activity of PpfA is modulated by the N terminus of a ParB analog (TlpT) [144]. In *C. crescentus*, another ParA homolog (MipZ) coordinates chromosome segregation with cell division by directly interfering with FtsZ polymerization [109]. MipZ binds DNA non-specifically and also interacts with ParB to create a bipolar protein gradient in the cells that restricts FtsZ ring formation to the mid cell, where the concentration of MipZ is lowest [95,109,145]. In *V. cholerae*, three ParA-like ATPases (ParA1, FlhG and ParC) interact with a polar transmembrane protein HubP to control polar localization of the chromosome origin, the chemotactic machinery and the flagellum [111]. These examples illustrate how diverse functions in biology can evolve from a general mechanism and are therefore interesting from both evolutionary and mechanistic standpoints.

Last but not least, a DNA segregation system that combines bacterial ParAB-like and eukaryotic histone-like components has been identified in the archaea *Sulfolobus* [146,147]. This system consists of an ATPase ParA, an atypical ParB adaptor and a novel centromere-binding protein AspA. The N-terminal domain of the archaeal ParB is similar to the bacterial ParB NTD; however, its C-terminal domain resembles an eukaryotic histone protein CenpA [146]. A long amino acid linker that connects the two domains of the archaeal ParB interacts with ParA, while its N-terminal domain binds AspA. AspA binds the centromere, thereby serves as a physical link between the archaeal ParA–ParB and the segregating DNA [146]. The hybrid nature of the archaeal DNA segregation machinery demonstrates how evolution has diversified DNA segregation systems, possibly to adapt to the specific needs of each organism, while keeping the general mechanism conserved across the three domains of life.

# 6. Final perspectives

Over 35 years of research has led to tremendous progress in understanding the molecular mechanism of the ParABS system and its roles in DNA segregation. Nevertheless, many mechanistic details are missing or only now starting to emerge. The recent discovery of cytidine triphosphate as a cofactor of ParB illustrates this point perfectly. Research with bacterial systems has already benefited tremendously from the recent explosion of interest and technological advances from the eukaryotic chromosome field. We predict that novel high–throughput sequencing-based methodologies, single-molecule imaging, single-molecule biophysics, and traditional biochemistry and genetics will continue to provide further insights into the mechanisms of chromosome segregation in bacteria. Finally, various orthogonal ParB–*parS* systems have been exploited to label and image DNA loci *in vivo*, in both bacteria and eukaryotes [148–151]. Recent studies have also expanded the utilization of the ParABS system in synthetic biology, for example, as part of a genetic circuit to enable asymmetric cell division in *E. coli* [152,153]. Such exciting developments will benefit from ongoing research into the mechanistic details of the ParABS system and its evolvability to acquire new functions.

**Data accessibility.** This article does not contain any additional data.
**Authors' contributions.** Both authors have contributed to the writing of the manuscript.
**Competing interests.** We declare we have no competing interests.
**Funding.** This study was funded by the Royal Society University Research Fellowship no. (UF140053) to T.B.K.L. and the Royal Society Research grant no. (RG150448) to A.S.B.J and T.B.K.L.

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
