## [Reviewer comments · Open Biology]

Review History

RSOB-20-0097.R0 (Original submission)

Review form: Reviewer 1

Recommendation

Accept with minor revision (please list in comments)

Do you have any ethical concerns with this paper?

No

Comments to the Author

Review by Jalal and Le comprehensively discusses current models of bacterial chromosome segregation. The manuscript is clearly organised and well written. I appreciate recalling the experimental data that support discussed models. However, since now manuscript is most focused on ParB I would suggest adding some additional information concerning ParA and SMC. Information on ParA structure and more details on the interactions with DNA would be useful. The mention of the current model of SMC action could be included. Finally, the short description of ParA homologues and their varied function would fit nicely in the chapter on the evolution of ParAB system.

Review form: Reviewer 2

Recommendation

Accept with minor revision (please list in comments)

Do you have any ethical concerns with this paper?

No

Comments to the Author

This review is very well written. It clearly summarizes the current literature on the ParABS system, which is broadly conserved in bacteria and is involved in plasmid and chromosome segregation. The review first discusses the current models for the formation and the role of the ParB-parS nucleoprotein complex. The authors describe each model and raise the concerns and remaining investigations for each one. In the following sections, the authors describe the role of the ParA protein in chromosome segregation, the link between chromosome segregation and chromosome organization (through the SMC protein) and finally discuss a few divergent ParABS system. In each section of this review, the models described by the authors are more centered on the general mechanisms shared between different species than on species-specific details. I think this review is a very good resource that, in my opinion, is valuable to a broad scientific community.

Comments:

lines 159-160: I would specify that the structure is lacking the C-terminal domain, especially because at lines 102-104 the authors state that no full length structure of ParB has been resolved yet.

line 204-206: this sentence has no verb: "A co-crystal structure showing CDP binding to the arginine-rich patch at the NTD of *B. subtilis* ParB (CTP was hydrolyzed to CDP during crystallization)"

line 211-212: "The *B. subtilis* ParB clamp can self-load at parS, without the need of a dedicated loading factor, and spreads by sliding to the neighboring DNA while still entrapping DNA". This sentence is a bit misleading. The sentence before this one states that CTP binding induces a clamp-like ParB. I think that the fact that CTP binding reduces ParB affinity for parS and induces the spreading along the DNA is not clear in this sentence and confuses the reader through the following paragraph.

line 238: in this section 2.6, it remained a little unclear to me if the different models described are totally disconnected or can be integrated together. It seems like the models are discussed here from the simplest 1D model to the most complex caging/CTP-binding model. Based on Figure 1, it also seems like the model are strongly connected but it's not really discussed in this section.

line 271: segregation "OF" the origin-proximal

lines 333-335: "mutations at the CTP-binding pocket of a ParB-like protein PadC were shown to impair PadC-ParA binding in vitro and gave rise to aberrant ParA localization patterns in vivo [51]." Maybe the authors could make it clearer here that CTP-bound PadC has more affinity to ParA or ParA preferentially binds to PadC-CTP.

line 416-417: "A mutational event that resulting in the grafting of an amphipathic helix", replace "that resulting" with "that resulted"

Decision letter (RSOB-20-0097.R0)

18-May-2020

Dear Dr Le

We are pleased to inform you that your manuscript RSOB-20-0097 entitled "Bacterial chromosome segregation by the ParABS system" has been accepted by the Editor for publication in Open Biology. The reviewer(s) have recommended publication, but also suggest some minor revisions to your manuscript. Therefore, we invite you to respond to the reviewer(s)' comments and revise your manuscript.

Please submit the revised version of your manuscript within 7 days. If you do not think you will be able to meet this date please let us know and we can extend this deadline for you.

- 1) A text file of the manuscript (doc, txt, rtf or tex), including the references, tables (including captions) and figure captions. Please remove any tracked changes from the text before submission. PDF files are not an accepted format for the "Main Document".
- 2) A separate electronic file of each figure (tiff, EPS or print-quality PDF preferred). The format should be produced directly from original creation package, or original software format. Please note that PowerPoint files are not accepted.
- 3) Electronic supplementary material: this should be contained in a separate file from the main text and meet our ESM criteria (see <http://royalsocietypublishing.org/instructions-authors#question5>). All supplementary materials accompanying an accepted article will be treated as in their final form. They will be published alongside the paper on the journal website and posted on the online figshare repository. Files on figshare will be made available approximately one week before the accompanying article so that the supplementary material can be attributed a unique DOI.

Online supplementary material will also carry the title and description provided during submission, so please ensure these are accurate and informative. Note that the Royal Society will not edit or typeset supplementary material and it will be hosted as provided. Please ensure that the supplementary material includes the paper details (authors, title, journal name, article DOI). Your article DOI will be 10.1098/rsob.2016[last 4 digits of e.g. 10.1098/rsob.20160049].

4) A media summary: a short non-technical summary (up to 100 words) of the key findings/importance of your manuscript. Please try to write in simple English, avoid jargon, explain the importance of the topic, outline the main implications and describe why this topic is newsworthy.

Images

Data-Sharing

It is a condition of publication that data supporting your paper are made available. Data should be made available either in the electronic supplementary material or through an appropriate repository. Details of how to access data should be included in your paper. Please see <http://royalsocietypublishing.org/site/authors/policy.xhtml#question6> for more details.

Data accessibility section

Sincerely,

The Open Biology Team

<mailto:openbiology@royalsociety.org>

Reviewer(s)' Comments to Author:

Referee: 1

Comments to the Author(s)

Review by Jalal and Le comprehensively discusses current models of bacterial chromosome segregation. The manuscript is clearly organised and well written. I appreciate recalling the experimental data that support discussed models. However, since now manuscript is most focused on ParB I would suggest adding some additional information concerning ParA and SMC. Information on ParA structure and more details on the interactions with DNA would be useful. The mention of the current model of SMC action could be included. Finally, the short description of ParA homologues and their varied function would fit nicely in the chapter on the evolution of ParAB system.

Referee: 2

Comments to the Author(s)

This review is very well written. It clearly summarizes the current literature on the ParABS system, which is broadly conserved in bacteria and is involved in plasmid and chromosome segregation. The review first discusses the current models for the formation and the role of the ParB-parS nucleoprotein complex. The authors describe each model and raise the concerns and remaining investigations for each one. In the following sections, the authors describe the role of the ParA protein in chromosome segregation, the link between chromosome segregation and chromosome organization (through the SMC protein) and finally discuss a few divergent ParABS system. In each section of this review, the models described by the authors are more centered on

the general mechanisms shared between different species than on species-specific details. I think this review is a very good resource that, in my opinion, is valuable to a broad scientific community.

Comments:

lines 159-160: I would specify that the structure is lacking the C-terminal domain, especially because at lines 102-104 the authors state that no full length structure of ParB has been resolved yet.

line 204-206: this sentence has no verb: "A co-crystal structure showing CDP binding to the arginine-rich patch at the NTD of *B. subtilis* ParB (CTP was hydrolyzed to CDP during crystallization)"

line 211-212: "The *B. subtilis* ParB clamp can self-load at parS, without the need of a dedicated loading factor, and spreads by sliding to the neighboring DNA while still entrapping DNA". This sentence is a bit misleading. The sentence before this one states that CTP binding induces a clamp-like ParB. I think that the fact that CTP binding reduces ParB affinity for parS and induces the spreading along the DNA is not clear in this sentence and confuses the reader through the following paragraph.

line 238: in this section 2.6, it remained a little unclear to me if the different models described are totally disconnected or can be integrated together. It seems like the models are discussed here from the simplest 1D model to the most complex caging/CTP-binding model. Based on Figure 1, it also seems like the model are strongly connected but it's not really discussed in this section.

line 271: segregation "OF" the origin-proximal

lines 333-335: "mutations at the CTP-binding pocket of a ParB-like protein PadC were shown to impair PadC-ParA binding in vitro and gave rise to aberrant ParA localization patterns in vivo [51]." Maybe the authors could make it clearer here that CTP-bound PadC has more affinity to ParA or ParA preferentially binds to PadC-CTP.

line 416-417: "A mutational event that resulting in the grafting of an amphipathic helix", replace "that resulting" with "that resulted"

Author's Response to Decision Letter for (RSOB-20-0097.R0)

See Appendix A.

Decision letter (RSOB-20-0097.R1)

21-May-2020

Dear Dr Le

We are pleased to inform you that your manuscript entitled "Bacterial chromosome segregation by the ParABS system" has been accepted by the Editor for publication in Open Biology.

Sincerely,
The Open Biology Team
mailto: openbiology@royalsociety.org

Appendix A

Thank you very much for the comments on our manuscript. We are very grateful to the editor and all reviewers for their critical and supportive comments. We have now revised the manuscript accordingly. Detailed responses to the specific points that reviewers have raised are given in the “response to referees” file.

Referee: 2

This review is very well written. It clearly summarizes the current literature on the ParABS system, which is broadly conserved in bacteria and is involved in plasmid and chromosome segregation. The review first discusses the current models for the formation and the role of the ParB-parS nucleoprotein complex. The authors describe each model and raise the concerns and remaining investigations for each one. In the following sections, the authors describe the role of the ParA protein in chromosome segregation, the link between chromosome segregation and chromosome organization (through the SMC protein) and finally discuss a few divergent ParABS system. In each section of this review, the models described by the authors are more centered on the general mechanisms shared between different species than on species-specific details. I think this review is a very good resource that, in my opinion, is valuable to a broad scientific community.

Comments:

lines 159-160: I would specify that the structure is lacking the C-terminal domain, especially because at lines 102-104 the authors state that no full-length structure of ParB has been resolved yet.

We thank the reviewer for pointing this out and we have now corrected this sentence.

line 204-206: this sentence has no verb: "A co-crystal structure showing CDP binding to the arginine-rich patch at the NTD of *B. subtilis* ParB (CTP was hydrolyzed to CDP during crystallization)"

We have now corrected this sentence to: "*A co-crystal structure showed CDP binding to the arginine-rich patch at the NTD of *B. subtilis* ParB (CTP was hydrolyzed to CDP during crystallization)*"

line 211-212: "The *B. subtilis* ParB clamp can self-load at parS, without the need of a dedicated loading factor, and spreads by sliding to the neighboring DNA while still entrapping DNA". This sentence is a bit misleading. The sentence before this one states that CTP binding induces a clamp-like ParB. I think that the fact that CTP binding reduces ParB affinity for parS and induces the spreading along the DNA is not clear in this sentence and confuses the reader through the following paragraph.

We thank the reviewer for pointing this out. We have now re-arranged sentences to make the order of facts/observations more logical.

line 238: in this section 2.6, it remained a little unclear to me if the different models described are totally disconnected or can be integrated together. It seems like the models are discussed here from the simplest 1D model to the most complex caging/CTP-binding model. Based on Figure 1, it also seems like the model are strongly connected but it's not really discussed in this section.

We refrained from suggesting whether different models are totally disconnected or can be integrated together. Instead, we suggested in our original manuscript that "*It is too early to answer this question adequately, given that many mechanistic details are still missing.*" The main purpose of section 2.6 was to raise a few important questions for the field, if answered, they have the potential to reconcile all models.

line 271: segregation "OF" the origin-proximal

Corrected

lines 333-335: "mutations at the CTP-binding pocket of a ParB-like protein PadC were shown to impair PadC-ParA binding in vitro and gave rise to aberrant ParA localization patterns in vivo [51]." Maybe the authors could make it clearer here that CTP-bound PadC has more affinity to ParA or ParA preferentially binds to PadC-CTP.

We have now included the phrase "*ParA preferentially binds to PadC-CTP than to apo-PadC*" to make the sentence clearer.

line 416-417: "A mutational event that resulting in the grafting of an amphipathic helix", replace "that resulting" with "that resulted"

Corrected

Referee: 1

Review by Jalal and Le comprehensively discusses current models of bacterial chromosome segregation. The manuscript is clearly organised and well written. I appreciate recalling the experimental data that support discussed models. However, since now manuscript is most focused on ParB I would suggest adding some additional information concerning ParA and SMC.

Comments:

Information on ParA structure and more details on the interactions with DNA would be useful. We thank the reviewer for pointing this out, and we have now mentioned X-ray crystallographic and HDX-MS studies that revealed the homodimerization interface and the DNA-binding surface of ParA. We have also added appropriate references to these crystallographic studies in section 3.0. Nevertheless, we refrained from describing crystallographic details in-depth due to (i) space constraints, and (ii) our wish to convey the core principle of the ParABS system to the broad readership. I hope referee 1 will agree with our judgment here.

The mention of the current model of SMC action could be included.

Precisely how SMC translocates on the chromosome is not clear; several very speculative models have been proposed. Furthermore, SMC is not the main focus of this review, thus we refer the reader to an excellent recent review for an in-depth discussion of the current model(s) of SMC action.

Yatskevich S, Rhodes J, Nasmyth K. (2019) Organization of Chromosomal DNA by SMC Complexes. *Annu Rev Genet.* 53:445-482. doi: 10.1146/annurev-genet-112618-043633

Finally, the short description of ParA homologues and their varied function would fit nicely in the chapter on the evolution of ParAB system.

This is an excellent suggestion, we have now added a section on other ParA homologs (e.g. PpfA in *R. sphaeroides*, MipZ in *C. crescentus*, and ParA1, FlhG, ParC in *V. cholerae*) to section 5.